# HCV universal EHR prompt successfully increases screening, highlights potential disparities

Benjamin Hack[1☉*], Kavya Sanghavi[2‡], Sravya Gundapaneni[2,3‡], Stephen Fernandez[2‡], Justin Hughes[2‡], Sean Huang[2,4‡], Peter Basch[2,5‡], Allan Fong[2‡], Dawn Fishbein[2,6☉]

**1** Georgetown University School of Medicine, Washington, D.C., United States of America, **2** MedStar Health Research Institute, Viral Hepatitis Research, Washington, D.C., United States of America, **3** Ross University School of Medicine, Miramar, FL, United States of America, **4** Georgetown University School of Nursing, Department of Health Systems Administration, Washington, D.C., United States of America, **5** MedStar Institute for Innovation, Ambulatory Electronic Health Record and Information Technology Policy, Washington, D.C., United States of America, **6** MedStar Washington Hospital Center, Department of Infectious Diseases, Washington, D.C., United States of America

☉ These authors contributed equally to this work.
‡ KS, SG, SF, JH, SH, PB and AF also contributed equally to this work.
* bh654@georgetown.edu

**Data Availability Statement:** All relevant data are within the paper and its Supporting Information files.

## Abstract

### Background & objectives

Screening for hepatitis C virus is the first critical decision point for preventing morbidity and mortality from HCV cirrhosis and hepatocellular carcinoma and will ultimately contribute to global elimination of a curable disease. This study aims to portray the changes over time in HCV screening rates and the screened population characteristics following the 2020 implementation of an electronic health record (EHR) alert for universal screening in the outpatient setting in a large healthcare system in the US mid-Atlantic region.

### Methods

Data was abstracted from the EHR on all outpatients from 1/1/2017 through 10/31/2021, including individual demographics and their HCV antibody (Ab) screening dates. For a limited period centered on the implementation of the HCV alert, mixed effects multivariable regression analyses were performed to compare the timeline and characteristics of those screened and un-screened. The final models included socio-demographic covariates of interest, time period (pre/post) and an interaction term between time period and sex. We also examined a model with time as a monthly variable to look at the potential impact of COVID-19 on screening for HCV.

### Results

Absolute number of screens and screening rate increased by 103% and 62%, respectively, after adopting the universal EHR alert. Patients with Medicaid were more likely to be screened than private insurance (OR$_{adj}$ 1.10, 95% CI: 1.05, 1.15), while those with Medicare

**Funding:** This work was funded by a grant from Gilead Sciences, Inc (https://www.gilead.com/). The funders had no role in study design, data collection and analysis, decision to publish, or preparation of the manuscript.

**Competing interests:** Dawn Fishbein, MD receives grant funding for this research from Gilead Sciences, Inc through her employer. She has stock ownership in Abbvie and Merck is a discretionary portfolio with a value less than $10,000. This does not alter our adherence to PLOS ONE policies on sharing data and materials.

were less likely ($OR_{adj}$ 0.62, 95% CI: 0.62, 0.65); and Black ($OR_{adj}$ 1.59, 95% CI: 1.53, 1.64) race more than White.

## Conclusions

Implementation of universal EHR alerts could prove to be a critical next step in HCV elimination. Those with Medicare and Medicaid insurance were not screened proportionately to the national prevalence of HCV in these populations. Our findings support increased screening and re-testing efforts for those at high risk of HCV.

## Introduction

Over the past two decades, viral hepatitis has become the foremost cause of global infectious disease related mortality, surpassing even HIV and tuberculosis [1]. The rapid increase in mortality, despite the availability of curative treatments, led international health leaders to develop action plans for the elimination of viral hepatitis [1]. Specifically, the World Health Organization presented the following aims regarding all hepatitis viruses by the year 2030: a 90% reduction in incidence, a 65% reduction in mortality, a 90% rate of diagnosing chronic cases, and an 80% treatment rate of individuals with chronic hepatitis [1].

An estimated 48% of global hepatitis-related mortality is attributable to hepatitis C virus infection, although a vast majority–between 75–90%–of hepatitis burden in the western hemisphere is HCV-related [1]. The approximate prevalence of HCV infection in the United States population is 1.7% (4.1 million people), based on 2013–2016 antibody (Ab) testing, which indicates either past or current infection [2]. The highest US prevalence was previously among those belonging to the Birth Cohort (BC), individuals born between 1945–1965, and the greatest single risk factor for HCV infection is current or historical intravenous drug use (IVDU) [3]. This led to guidelines which recommended one-time screening for the BC, more often for patients with risk factors such as IVDU, incarceration or substance use disorders [3]. However, based on cost-benefit analyses and the rising prevalence of HCV in younger populations, the guidelines were updated in March 2020 to recommend one-time universal HCV screening for all adults between the ages of 18–79, all pregnant women, and continued testing for those with risk factors [4–8]. This marked an important step in the elimination of HCV in the USA.

One goal under elimination is "micro-elimination", as in eliminating HCV in a given population; however, the first challenge of this aim is developing a strategy for improving screening efforts [5]. With the advent of universal screening, the healthcare provider still needs to recognize which patients require their screening, and if re-testing is appropriate based on risk factors [7, 8]. Fortunately, clinical decision support (CDS) alerts built into electronic health records (EHR) may lessen the burden of this responsibility [9–14]. Additionally, such alerts may improve screening rates and, ultimately, connection to care [14].

In the studied large academic healthcare system, with approximately 2.7 million outpatient visits annually, HCV screening CDS EHR alerts have reflected the changing recommendations over the past decade. A system-wide EHR alert for screening the BC began in 2015, was temporarily discontinued by January, 2017 during transition from one EHR to another, and then re-implemented in September, 2018. In May, 2020, less than two months after the update in US Preventive Services Task Force (USPSTF) and Centers for Disease Control (CDC) recommendations, a universal HCV screening CDS alert for eligible patients was adopted. This observational study aims to describe the changes over time in HCV screening rates and population

characteristics in the period preceding and following implementation of an EHR alert across the health system, and to use this information along with comparisons to the unscreened population to suggest future avenues for HCV micro-elimination.

## Materials and methods

### Extraction and variables

Data was extracted from the EHR for patients who received an HCV antibody (HCVAb) screen between January 2017 and October 2021 and were seen in the outpatient setting. Data was extracted by biomedical informatics and provided to the team in an Excel or csv database. Variables included age, sex (female, male, other/unknown), race (White, Black, Asian, American Indian or Alaska Native, Native Hawaiian or Pacific Islander, Multiple, Other/Unknown), ethnicity (non-Hispanic, Hispanic, Other/Unknown), address including state and zip code, insurance name and type (government, Private, Medicare, Medicaid, Self-Pay, Other/ unknown), and date of antibody test.

A separate data set was extracted for all outpatient visits during the period January 1, 2020 and October 31, 2020, which included all variables above as well as whether the patient received an HCV Ab screen and the dates of the screenings, in addition to all outpatient visit dates. If any of the screening dates were 31 days before or after an appointment date, then the patient was considered screened for that appointment date. This measure was taken to account for providers ordering tests to be completed shortly before or after scheduled outpatient appointments. If a patient's closest appointment was before the alert implementation and the screening date was after, then the date of their screen was considered pre-alert to provide a more conservative estimate of the alert's effect.

Consent was waived for participants in our study as all data was deidentified. This study was approved by the MedStar Health Research Institute Institutional Review Board.

### Statistical analysis

We used frequencies and percentages to summarize the distribution of our categorical variables; and means with standard deviations or medians with interquartile ranges to summarize our continuous variables. To account for repeated measures of patients who had outpatient visits pre-alert and post-alert, we conducted bivariate and multivariable mixed effects logistic regression analyses with screened for HCV as the dependent categorical outcome variable. We evaluated two different models by adding an interaction term between time period and sex, and an interaction term between time period and race to examine if the relationship between the independent variable and the outcome was affected by time period. Any statistically significant interaction term was included in the final model. Our final models included socio-demographic covariates of interest (i.e. age, race, ethnicity, sex and insurance type), time period (pre/post) and an interaction term between time period and sex. We also examined a model with time as a monthly variable to look at the potential impact of COVID-19 on screening for HCV. We evaluated the inclusion of random effects versus a linear regression model using the likelihood ratio test. Statistical analyses were conducted on Stata version 17 (StataCorp LLC). Significance level was set at 0.05.

## Results

### Absolute screening numbers and characteristics 1/2017-10/2021

During the study period, there were 155,012 HCV Ab screen recipients in outpatient locations. Between January 2017 and May 2020 (pre-alert), 84,139 screens were completed at a rate of

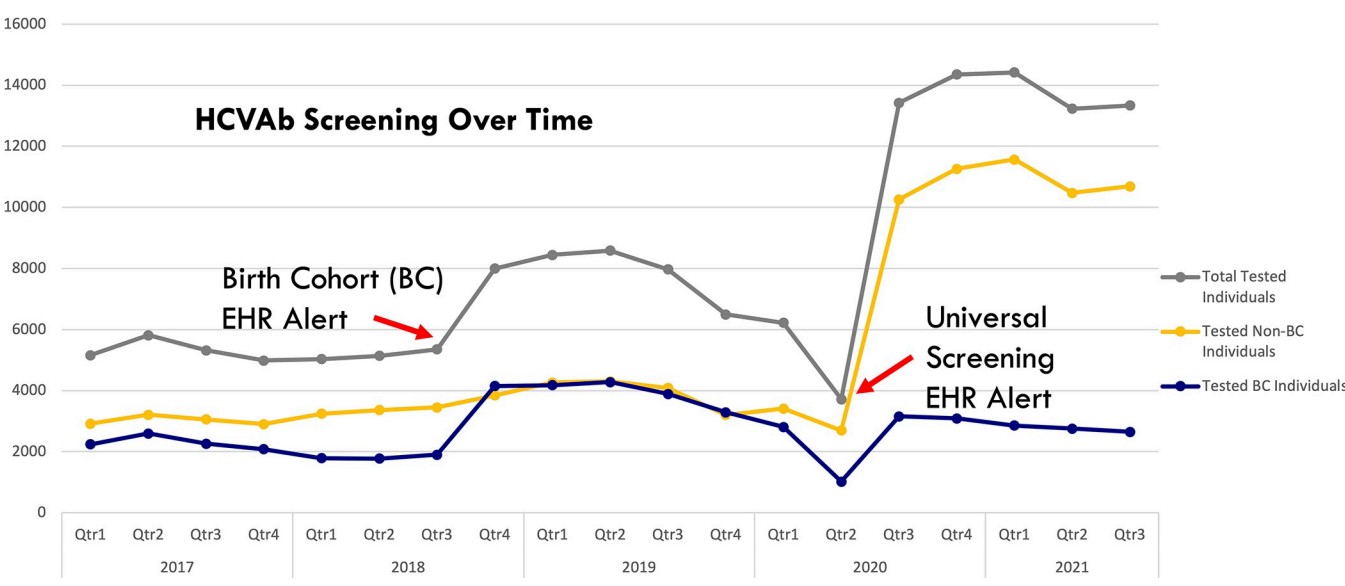

**Fig 1. HCV antibody screening over time.** Absolute HCV antibody screens, birth cohort (BC) vs. non-BC over time January 2017 to October 2021 for all MedStar outpatient sites. The first arrow (left) indicates the approximate re-implementation of the electronic health record (EHR) alert for the birth cohort in September 2018. The second arrow (right) indicates the approximate implementation of the universal screening EHR alert on May 28th, 2020.

2,052 screens per month. Between June 2020 and October 2021 (post-alert), 70,873 screens were completed at a rate of 4,169 screens per month, representing a 103% increase in monthly screening rate (Fig 1). The mean age of HCV Ab screening recipients decreased from 53.8 in 2017 to 45.1 in 2021 (Table 1). The percent of HCV Ab screen recipients who did not belong to the BC (non-BC) increased from 56.8% in 2017 to 79.8% in 2021 of the total screened.

In the year prior to re-implementation of the system BC EHR alert for HCV screening in September 2018 (October 2017 –September, 2018), 20,514 screens were completed at a rate of 1,710 screens per month; the year following implementation (October 2018 –September 2019), 33,001 screens were completed at a rate of 2,750 screens per month. This represented a 60.9% increase in the mean monthly screening rate. The mean age of those screened from prior to the BC alert was 50.7 years, which increased to a mean age of 53.9 in the year after the BC alert; prior to the BC alert, 36.8% of all screened belonged to the BC; the year after, 50.0% of those screened were in the BC (Fig 1, Table 1).

Over the entire study period, 66.2% of all screened with an indicated sex (n = 146,564) were female; there was an absolute increase from 63.3% in 2017 to 67.7% in 2021. The two most prominent races of those with an identified race (n = 137,331) were White and African American/Black, which comprised 45.7% and 47.5% of the sample respectively. During 2017, 44.2% of those screened were White and 49.5% were African-American/Black; in 2021, 45.7% were White and 47.2% were African American/Black (Table 1).

Overall, 64.0% of the screened individuals with an indicated insurance type (n = 137,061) had private/commercial insurance, 11.9% Medicare, and 21.3% Medicaid. In 2017, 63.7% of those screened had private insurance which increased to 65.3% by 2021. In 2017, 11.4% of those screened had Medicare, which increased to 14.6% in 2020, then decreased to 10.5% in 2021 (Table 1).

Overall, 72.3% of those screened who indicated a state of residence (n = 152,107) lived in Maryland, 22.0% in Washington, District of Columbia and 5.7% in Virginia. In 2017, 66.9% of those screened lived in Maryland, which increased to 76.1% in 2019, then decreased to 70.6% in 2021.

**Table 1. Absolute numbers and characteristics of HCV screened individuals from 1/2017-10/2021.**

| | 2017 | | 2018 | | 2019 | | 2020 | | 2021 | | Total | |
|---|---|---|---|---|---|---|---|---|---|---|---|---|
| | Absolute | Percent | Absolute | Percent | Absolute | Percent | Absolute | Percent | Absolute | Percent | Absolute | Percent |
| Total Screens | 21,280 | -- | 23,524 | -- | 31,500 | -- | 37,718 | -- | 40,990 | -- | 155,012 | -- |
| Age (mean) | 53.8 | -- | 51.7 | -- | 53.7 | -- | 47.3 | -- | 45.1 | -- | 49.6 | -- |
| Non-BC | 12,096 | 56.8 | 13,909 | 59.1 | 15,864 | 50.4 | 27,634 | 73.2 | 10,084 | 79.8 | 102,228 | 65.9 |
| BC | 9,184 | 43.2 | 9,615 | 40.9 | 15,636 | 49.6 | 10,084 | 26.7 | 8,265 | 20.2 | 52,784 | 34.1 |
| Sex | | | | | | | | | | | | |
| Male | 7,518 | 36.7 | 7,819 | 34.7 | 10,487 | 35.1 | 11,204 | 32.0 | 12,491 | 32.3 | 49,519 | 33.8 |
| Female | 12,967 | 63.3 | 14,725 | 65.3 | 19,425 | 64.9 | 23,782 | 68.0 | 26,146 | 67.7 | 97,045 | 66.2 |
| Race/Ethnicity | | | | | | | | | | | | |
| Black* | 9,374 | 49.3 | 10,471 | 50.0 | 12,516 | 44.5 | 15,451 | 47.3 | 16,955 | 47.3 | 64,767 | 47.4 |
| White | 8,375 | 44.0 | 8,854 | 42.3 | 13,267 | 47.2 | 15,406 | 47.1 | 16,444 | 45.8 | 62,346 | 45.6 |
| Hispanic/LA** | 880 | 4.3 | 981 | 4.4 | 1089 | 3.7 | 910 | 2.6 | 1,250 | 3.3 | 5,110 | 3.5 |
| Asian | 544 | 2.9 | 713 | 3.4 | 868 | 3.1 | 1162 | 3.6 | 1,619 | 4.5 | 4,906 | 3.6 |
| Insurance | | | | | | | | | | | | |
| Private | 11,747 | 63.7 | 13,552 | 64.7 | 18,004 | 64.3 | 20,870 | 62.2 | 23,560 | 65.3 | 87,733 | 64.0 |
| Medicare | 2,111 | 11.4 | 1,960 | 9.4 | 3,538 | 12.6 | 4,908 | 14.6 | 3,805 | 10.5 | 16,322 | 11.9 |
| Medicaid | 3,957 | 21.4 | 4,731 | 22.6 | 5,814 | 20.8 | 7,070 | 21.1 | 7,594 | 21.1 | 29,166 | 21.3 |
| Location | | | | | | | | | | | | |
| MD | 13,815 | 66.9 | 16,491 | 72.0 | 23,538 | 76.2 | 27,575 | 74.2 | 28,578 | 70.6 | 109,987 | 72.3 |
| DC | 5,462 | 26.4 | 4,939 | 21.6 | 5,695 | 18.4 | 7,553 | 20.3 | 9,758 | 24.1 | 33,407 | 22.0 |
| VA | 1,390 | 6.7 | 1,474 | 6.4 | 1,672 | 5.4 | 2,040 | 5.5 | 2,137 | 5.3 | 8,713 | 5.7 |

Percent is equal to the percent of HCV screened patients with a given characteristic over the total screened with the given characteristic.

*Black race includes all individuals who indicated their race as 'Black,' 'African American' or 'Black or African American.'

**Hispanic and Latin American represents all those with ethnicities listed as Hispanic, Latin American, Cuban, Dominican, Puerto Rican, South American, or Colombian; race and ethnicity are collected separately, therefore Hispanic/LA is not independent from the races listed.

## Screening rates per visit and demographic comparisons before and after HCV universal alert, January, 2020—October, 2020

To model the effect of the universal screening alert on the screening rate per outpatient visit and on the characteristics of those screened, a symmetric time period was analyzed which centered around the month of the implementation date. Absolute number and percentage of characteristics for the entire unique outpatient and the HCV screened populations from 1/1/2020-10/31/2020 are reported in the Table 2. The outcomes of a mixed effect multivariable analysis model are presented in Table 3. Adjusting for all other covariates, there was a 62% increased odds of being screened after the universal alert (post-alert) compared to prior to the universal alert (pre-alert) ($OR_{adj}$ 1.62, 95% CI: 1.54, 1.70; Table 3). Compared to January 2020, there was a statistically significantly decreased odds of screening in March–May 2020, no significant difference in June 2020, and then increased odds of screening thereafter (Table 3, Fig 2). March–May 2020 also represents a major clinic shutdown during the COVID-19 pandemic.

On multivariable analysis, there was a statistically significant interaction between sex and time period–females showed a 10%-decreased odds of being screened post-alert compared to male patients pre-alert ($OR_{adj}$ 0.90, 95% CI: 0.85, 0.95). Compared to patients who identified as White, patients that identified as Black, or any other race, were associated with a 59% ($OR_{adj}$ 1.59, 95% CI: 1.53, 1.64) and 61% increased odds ($OR_{adj}$ 1.61 95% CI: 1.52, 1.71), respectively, of being screened for HCV. Each one-year increase in age was associated with a 2% decreased

**Table 2. Characteristics and percentages of all individuals (screened and unscreened) and all screened individuals seen in the MedStar outpatient setting for 10 months between 1/1/2020-10/31/2020.**

| | All Pre-Alert** (n = 429,085) | | All Post-Alert** (n = 501,076) | | Screened Pre-Alert** (n = 10,717) | | Screened Post-Alert** (n = 21,295) | |
|---|---|---|---|---|---|---|---|---|
| | N or Median | %/IQR | N or Median | % or IQR | N or Median | % or IQR | N or Median | % or IQR |
| Age | 59 | 41–73 | 59 | 41–73 | 49 | 29–69 | 48 | 28–68 |
| Sex | | | | | | | | |
| Male | 169,470 | 39.50% | 198,082 | 39.53% | 3,371 | 31.45% | 6,789 | 31.88% |
| Female | 259,556 | 60.49% | 302,897 | 60.45% | 7,343 | 68.52% | 14,503 | 68.11% |
| Race | | | | | | | | |
| White | 217,342 | 50.65% | 258,498 | 51.59% | 4,371 | 40.79% | 8,698 | 40.85% |
| Black/African American | 141,955 | 33.08% | 156,288 | 31.19% | 4,612 | 43.03% | 9,044 | 42.47% |
| American Indian/Alaska Native | 1,036 | 0.24% | 1,164 | 0.23% | 33 | 0.31% | 73 | 0.34% |
| Asian | 10,022 | 2.34% | 11,241 | 2.24% | 331 | 3.09% | 739 | 3.47% |
| Native Hawaiian/Pacific Islander | 204 | 0.05% | 249 | 0.05% | 10 | 0.09% | 17 | 0.08% |
| Multiple | 2,947 | 0.69% | 3,406 | 0.68% | 84 | 0.78% | 174 | 0.82% |
| Other | 27,615 | 6.44% | 31,905 | 6.37% | 790 | 7.37% | 1,579 | 7.41% |
| Unknown/Declined | 27,964 | 6.52% | 38,325 | 7.65% | 486 | 4.53% | 971 | 4.56% |
| Ethnicity | | | | | | | | |
| Non-Hispanic | 379,089 | 88.35% | 437,714 | 87.35% | 9,693 | 90.45% | 19,227 | 90.29% |
| Hispanic | 10,174 | 2.37% | 11,314 | 2.26% | 290 | 2.71% | 659 | 3.09% |
| Other/Unknown | 39,822 | 9.28% | 52,048 | 10.39% | 734 | 6.85% | 1,409 | 6.62% |
| Insurance Category | | | | | | | | |
| Government | 4,542 | 1.06% | 5,231 | 1.04% | 92 | 0.86% | 205 | 0.96% |
| Medicaid | 61,786 | 14.40% | 70,520 | 14.07% | 2,282 | 21.29% | 4,245 | 19.93% |
| Medicare | 115,163 | 26.84% | 124,708 | 24.89% | 1,961 | 18.30% | 3,264 | 15.33% |
| Private | 187,298 | 43.66% | 225,650 | 45.03% | 5,225 | 48.75% | 11,454 | 53.79% |
| Self-Pay | 13,150 | 3.06% | 19,500 | 3.89% | 243 | 2.27% | 414 | 1.94% |
| Other/Unknown | 47,146 | 10.99% | 55,467 | 11.07% | 914 | 8.53% | 1,713 | 8.04% |

** Pre-alert: Jan 1st 2020 –May 31st 2020; Post-alert: June 1st 2020 –Oct 31st 2020.

odds of being screened for HCV ($OR_{adj}$ 0.98, 95% CI: 0.97, 0.98). Compared to patients who had commercial or private insurance, patients who had Medicaid were associated with an increase in odds of screening ($OR_{adj}$ 1.10, 95% CI: 1.05, 1.15) whereas patients that had government ($OR_{adj}$ 0.83, 95% CI: 0.71, 0.97), or Medicare ($OR_{adj}$ 0.62, 95% CI: 0.62, 0.65) were all associated with a decrease in odds of screening (Table 3).

## Discussion

### Implementation of the HCV EHR alert and screening rates

The data presented here suggests that implementation of an EHR alert for HCV universal screening in a large health system is an effective strategy for accomplishing increased screening rates. Absolute number of screens per month was 2,052 before the universal screening guideline and EHR alert were adopted, and this rate increased by 103% to 4,169 screens soon after the implementation of the EHR alert for universal screening at the end of May 2020, despite the COVID-19 pandemic. To determine whether it was a result of the change in USPSTF and CDC guideline recommendations alone or the addition of the alert which was associated with an increase in screening, screening rates among all outpatient appointments from a smaller

**Table 3. Multivariable mixed effects logistic regression models for pre-post 6/1/2020 (model one) and by month 1/2020-10/2020 (model two).**

| Adjusted b estimates for outcome | | | | | | | |
|---|---|---|---|---|---|---|---|
| **Random Effects Model 1 (n = 1,799,767, intercept = 0.00 [0.0,0.01], constant variance 4.17 (0.06))** | | | | **Random Effects Model 2 (n = 1,799,767, intercept = 0.01 [0.00, 0.01], Constant Variance 4.12 (0.06))** | | | |
| | | | | **Fixed Effects** | | | |
| Covariate | Coefficient | 95% CI | p-value | Covariate | Coefficient | 95% CI | p-value |
| Time period | | | | Time period | | | |
| Pre-alert | Ref | -- | -- | Jan 2020 | Ref | [1.01, 1.12] | **0.032** |
| Post-alert | 1.62 | [1.54, 1.70] | **<0.001** | Feb 2020 | 1.06 | [0.87, 0.98] | **0.010** |
| | | | | March 2020 | 0.93 | [0.44, 0.51] | **<0.001** |
| | | | | April 2020 | 0.47 | [0.55, 0.63] | **<0.001** |
| | | | | May 2020 | 0.59 | [0.95, 1.06] | 0.938 |
| | | | | June 2020 | 1.00 | [1.18, 1.31] | **<0.001** |
| | | | | July 2020 | 1.24 | [1.17, 1.29] | **<0.001** |
| | | | | Aug 2020 | 1.23 | [1.29, 1.43] | **<0.001** |
| | | | | Sept 2020 | 1.36 | [1.36, 1.51] | **<0.001** |
| | | | | Oct 2020 | 1.43 | [0.97, 0.98] | **<0.001** |
| Sex | | | | Sex | | | |
| Male | Ref | -- | -- | Male | Ref | -- | -- |
| Female | 1.26 | [1.20, 1.32] | **<0.001** | Female | 1.18 | [1.14, 1.22] | **<0.001** |
| Interaction term time*sex | | | | | | | |
| Pre*male | Ref | | | | | | |
| Post*female | 0.90 | [0.85, 0.95] | **<0.001** | | | | |
| Age | 0.98 | [0.97, 0.98] | **<0.001** | Age | 0.98 | [0.97, 0.98] | **<0.001** |
| Race | | | | Race | | | |
| White | Ref | -- | -- | White | Ref | -- | -- |
| Black/African American | 1.59 | [1.53, 1.64] | **<0.001** | Black/ African American | 1.58 | [1.53, 1.64] | **<0.001** |
| Other | 1.61 | [1.52, 1.71] | **<0.001** | Other | 1.60 | [1.52, 1.70] | **<0.001** |
| Ethnicity | | | | Ethnicity | | | |
| Non-Hispanic | Ref | -- | -- | Non-Hispanic | Ref | -- | -- |
| Hispanic | 1.08 | [0.97, 1.20] | 0.177 | Hispanic | 1.07 | [0.97, 1.19] | 0.180 |
| Insurance type | | | | Insurance type | | | |
| Private | Ref | -- | -- | Private | Ref | -- | -- |
| Government | 0.83 | [0.71, 0.97] | **0.022** | Government | 0.83 | [0.71, 0.98] | **0.023** |
| Medicaid | 1.10 | [1.05, 1.15] | **<0.001** | Medicaid | 1.11 | [1.06, 1.16] | **<0.001** |
| Medicare | 0.62 | [0.60, 0.65] | **<0.001** | Medicare | 0.63 | [0.60, 0.66] | **<0.001** |
| Self-Pay | 0.71 | [0.64, 0.78] | **<0.001** | Self-Pay | 0.71 | [0.65, 0.79] | **<0.001** |

time period (1/1/20–10/31/20) were analyzed. This allowed for a symmetric time range surrounding the implementation date and a month-by-month analysis. Screening rates rose precipitously after May 2020 and remained at levels significantly higher than those in January 2020. Overall, there was an 162% increase in screening rate in the five months after the alert compared to the five months before. This leads us to conclude that the EHR alert adopted on 5/28/2020 was associated with a significant increase in screening rate, although not the only likely factor contributing to the rise in screening rate. This increase was not seen between March and May, when the universal screening guidelines had been published, but the system EHR alert had not yet been adopted.

As aforementioned, in March 2020 major USA guideline publishing organizations recommended changing practice to one-time universal screening for HCV in all patients ages 18–79,

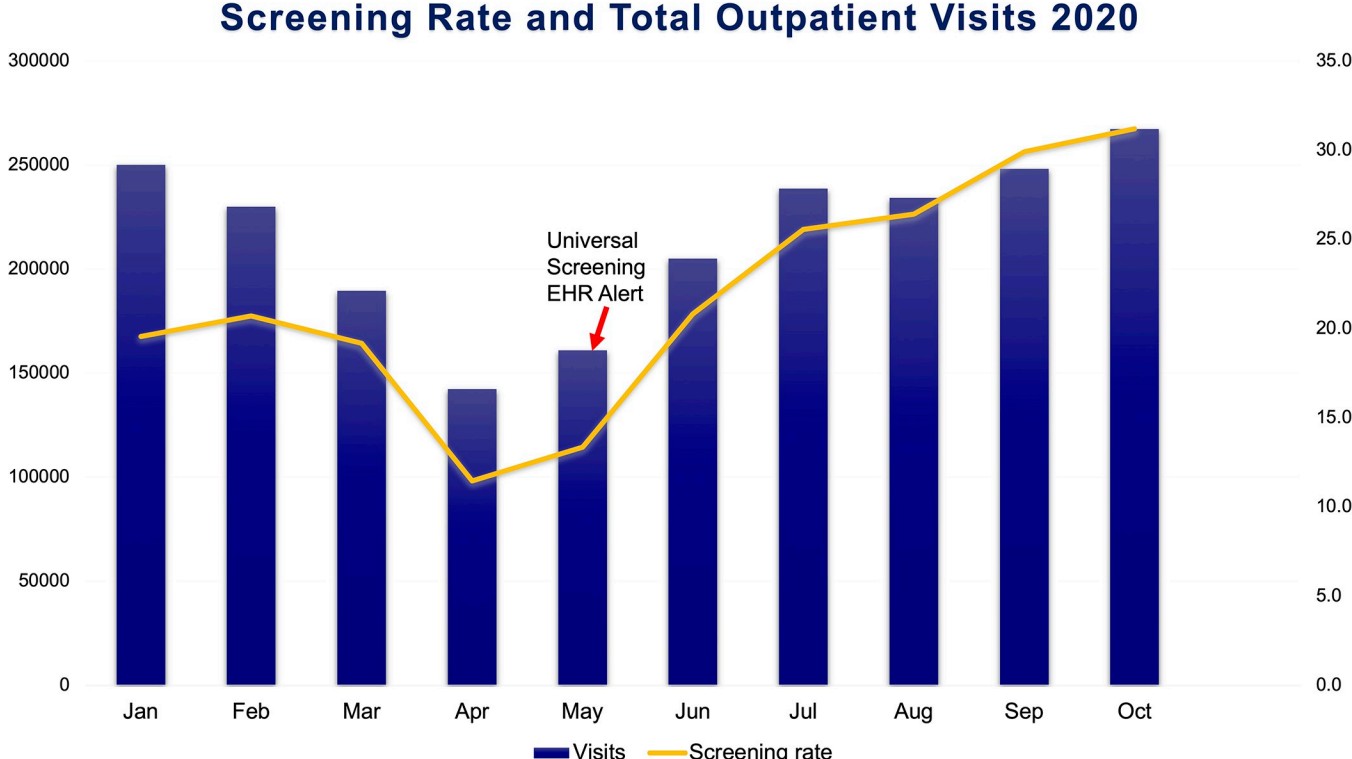

**Fig 2. Screening rate and total outpatient visits 2020.** Screening rate per 1,000 outpatient visits and total outpatient visits, both screened and unscreened, between January 1st, 2020 to October 31st, 2020. The arrow indicates the approximate implementation of the universal screening electronic health record alert on May 28th, 2020.

all pregnant women and with repeated testing in those with risk factors [7, 8]. Unfortunately, this drastic and promising change in elimination strategy coincided with the rise of the COVID-19 pandemic. In the month of March 2020, national outpatient visits began declining until they reached a 70% decline from the previous year in April 2020; afterwards, they began increasing again until they reached pre-COVID levels in August 2020 [15]. Our results are consistent with this finding, as absolute visits decreased rapidly between March and April; this could also explain the concurrent decrease in HCV screening rate, despite the coincidental adoption of universal screening guidelines. However, it does not likely explain the rapid increase in screening rates after May 2020, as an increase in the number of visits does not necessarily translate to an increase in how many individuals are screened per 1,000 visits. In addition, although the number of visits returned to pre-COVID levels in July 2020, screening rates were significantly higher than previously. These considerations support the conclusion that the EHR alert was driving the encouraging screening trends.

Even before the onset of the COVID-19 pandemic, the United States was not on track to achieve the elimination goals set out by the World Health Organization, according to recent modelling of HCV screening and connection to care [16]. Although global HCV prevalence decreased from 0.9% in 2015 to 0.7% in 2020, they estimated that 78% of HCV cases were undiagnosed [16]. This suggests that the first step of elimination, thoroughly screening the general population, may prove to be the most important. Thus, developing strategies such as EHR alerts to enhance population screening will play a significant role in HCV elimination.

There are several other reports in the literature of EHR prompts or alerts improving rates of screening, specifically for HCV [9–14, 17–19]. Most of the studies were completed when only

the BC screening guidelines were in place. Sidlow et al. (2015) reported a 254% increase in HCV screening after incorporating a clinical reminder into the EHR for primary care patients who belonged to the BC and still required a screen. Barter et al. (2021) systematically reviewed 11 studies and found that EHR utilization increased HCV screening rates across all of them. These findings are consistent with those presented here, as the re-implementation of the EHR alert for BC screening in September 2018 was followed by a 61% increase in absolute number of screens and 6.3% increase in age of those screened.

Only a few studies have observed screening data after the universal screening guidelines were published in March 2020 [17–19]. Of note, all of these studies were conducted in emergency departments. The largest of these studies looked at screening rates before and after implementation of a "Best Practice Advisory" (BPA) for universal screening; they found that 0.3% (218/69,604) of patients were screened for 11 months pre-BPA, and 22.0% (14,981/68,225) were screened in the 11 months post-BPA, representing a 73-fold increase in screening rate [17]. This increase is likely more drastic than the results found here as we measured screening rates in the outpatient setting, where baseline screening rates were already higher. Taking our findings and the previous literature into account, EHR alerts for universal screening, along with other clinical decision support tools, can play an important role in HCV elimination.

### Characteristics of screened and unscreened individuals: Who are we missing?

Regardless of universal screening and EHR alerts, the importance of identifying patients who are not getting screened cannot be understated. To understand this problem, we compared the basic characteristics of unique patients who were HCV screened to unique patients who were not screened by the universal alert over the 10-month period between 1/1/2020 and 10/31/2020 (pre- and post-alert). Multivariable fixed effects modeling revealed that patients identifying as female were 17% more likely to be screened than males, Black individuals 66% more likely than White, and Hispanic 16% more than non-Hispanic. According to national data from 2019, the majority of actively HCV-infected individuals identify as male (63.7%), thus representing one discrepancy between who is getting tested and who should be getting tested [20]. Older data found that the national prevalence of HCV was 1.0% among non-Hispanic White individuals and 2.3% among non-Hispanic Black individuals, suggesting that the higher screening rate of Black individuals may be warranted [21]. Consideration of these simple demographics in screening may help providers make clinical decisions while limiting bias.

A notable discrepancy we present here was that in insurance type. Patients with Medicare or Medicaid insurance were screened at a rate about 60% and 110%, respectively, of those with private insurance. One analysis of national data from 2001–2010 found that the prevalence of HCV in those with private insurance was 0.81%, while it was 1.24% in those with Medicare, and 2.58% in those with Medicaid [22]. This data indicates that HCV prevalence is approximately 300% higher in Medicaid patients compared to privately insured, while our data revealed only a 10% higher screening rate. Therefore, our results demonstrate that, although screening rates were slightly higher in patients with Medicaid than private insurance, they are not proportionate to the prevalence of HCV in these populations. Those with Medicare insurance were also tested at disproportionately low rates, though this could partially result from the increased screening of lower ages. Policies to improve screening could incorporate these disparities, perhaps with increased clinical decision support for those with Medicaid and Medicare insurance.

## Limitations

Many of the limitations of this study result from the vast size of the data sets and the lack of granularity on the unique patient level. Temporal observations of unique-screened patients over the 58-month period required us to only include the last appointment at which an individual was screened. This decision was made because a patient who is HCV Ab positive will always remain Ab positive, though a patient with a negative screen can later seroconvert and become positive. Although this could skew the results toward a later temporal distribution, a simple calculation of average screens per person was conducted to find that, among those screened, a vast majority were only screened once, with an average of approximately 1.3 screens per person. Therefore, the descriptive statistics presented here were unlikely affected by the choice of last versus first screening–this measure was not needed for the 10-month period analysis.

Our temporal analysis of unscreened versus screened individuals was limited by database software. The pre-post analysis for screening rates before and after the universal alert was conducted on a limited, symmetric time frame rather than the entire 58-month period. With approximately 2.7 million outpatient appointments annually, the size of the potential sample over a 58-month period exceeded the capacity of the database. Therefore, a five-month period preceding and following the alert implementation was chosen for a more statistically manageable, yet still large, sample size, as well as a historically meaningful period. For this analysis, univariable statistical comparison was not appropriate for pre-post analysis because individual patients could have been seen at multiple appointments before being screened; therefore, a multivariable mixed-effect analysis was completed. Additionally, the pre-post comparison of screening rate was possibly influenced by the low number of visits and screens at the beginning of the COVID-19 pandemic.

The size of the database also limited our ability to accurately collect and analyze certain variables. Given that pregnancy is a consideration in HCV screening, pregnancy status of participants could have impacted screening rates. Universal pregnancy screening was not recommended until June, 2021 by the American College of Gynecology nor the Society for Maternal Fetal Medicine, and thus this was not captured separately. Additionally, we were not able to differentiate whether patients' visits were at primary care or specialty clinic sites.

Another limitation of this study is that data before 2017 was not available to us, as the health system transitioned from one EHR to another. Thus, our analysis could not account for patients who were screened before 2017, and therefore would not necessarily need to be screened during this later time period unless they had continued risk factors.

One final limitation of this study is that we could not explore provider interaction with the screening alert, which prevented us from determining potential reasons patients went unscreened. With this data, we would have been able to observe which providers viewed the alert, and then how many indicated a reason for not screening the patient; for example, if the patient had previously been screened or treated for HCV in a different health system.

## Conclusions

Our findings support the universal adoption of an EHR prompt for universal HCV screening of adults 18–69. The absolute number and rate of HCV screening increased over time, and became dramatically higher with the implementation of the universal screening recommendation and EHR alert. Our findings also support screening and re-testing of individuals who are at higher risk of HCV infection. Females were tested more than males, despite data suggesting that males have a higher HCV prevalence. In addition, patients with Medicare or Medicaid insurance were not screened proportionately to the relative national prevalence of HCV in

these populations. Implementation of such practices could play a crucial role as the first step in identification and then elimination of HCV.

## Supporting information

**S1 Data.**
(XLSX)

**S2 Data.**
(ZIP)

## Author Contributions

**Conceptualization:** Benjamin Hack, Sravya Gundapaneni, Dawn Fishbein.

**Data curation:** Kavya Sanghavi, Stephen Fernandez, Justin Hughes, Allan Fong.

**Formal analysis:** Kavya Sanghavi.

**Funding acquisition:** Peter Basch, Dawn Fishbein.

**Investigation:** Benjamin Hack, Sravya Gundapaneni, Justin Hughes, Allan Fong.

**Methodology:** Stephen Fernandez, Justin Hughes, Allan Fong, Dawn Fishbein.

**Project administration:** Peter Basch, Dawn Fishbein.

**Resources:** Stephen Fernandez, Sean Huang, Peter Basch, Dawn Fishbein.

**Software:** Stephen Fernandez, Justin Hughes, Allan Fong.

**Supervision:** Sean Huang, Dawn Fishbein.

**Validation:** Kavya Sanghavi, Justin Hughes.

**Visualization:** Benjamin Hack, Dawn Fishbein.

**Writing – original draft:** Benjamin Hack, Dawn Fishbein.

**Writing – review & editing:** Benjamin Hack, Kavya Sanghavi, Sravya Gundapaneni, Sean Huang, Peter Basch, Dawn Fishbein.

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
