## [Decision Letter · Decision Letter 0]

1 Nov 2022

PONE-D-22-18564HCV Universal EHR Prompt Successfully Increases Screening, Highlights Potential DisparitiesPLOS ONE

Dear Dr. Hack,

Thank you for submitting your manuscript to PLOS ONE. After careful consideration, we feel that it has merit but does not fully meet PLOS ONE’s publication criteria as it currently stands. Therefore, we invite you to submit a revised version of the manuscript that addresses the points raised during the review process.

Both Reviewer #1 and I feel that this paper requires only minor revisions.  As we were unable to get a second reviewer in a timely manner for this manuscript, I undertook the second review in my capacity as Academic Editor.I believe that Reviewer #1 comments capture the revisions needed. In addition, I suggest the following minor edits:1. Please include a reference for micro-elimination (Lines 75-76).2. Lines 136-137:  I realize authors may be rounding, however, I think the text and table should be the same- is the percent 79.8% or 80%?Please submit your revised manuscript by December 15, 2022. If you will need more time than this to complete your revisions, please reply to this message or contact the journal office at plosone@plos.org. Please include the following items when submitting your revised manuscript:A rebuttal letter that responds to each point raised by the academic editor and reviewer(s). You should upload this letter as a separate file labeled 'Response to Reviewers'.A marked-up copy of your manuscript that highlights changes made to the original version. You should upload this as a separate file labeled 'Revised Manuscript with Track Changes'.An unmarked version of your revised paper without tracked changes. You should upload this as a separate file labeled 'Manuscript'.If applicable, we recommend that you deposit your laboratory protocols in protocols.io to enhance the reproducibility of your results. Protocols.io assigns your protocol its own identifier (DOI) so that it can be cited independently in the future. For instructions see: https://journals.plos.org/plosone/s/submission-guidelines#loc-laboratory-protocols. Additionally, PLOS ONE offers an option for publishing peer-reviewed Lab Protocol articles, which describe protocols hosted on protocols.io. Read more information on sharing protocols at https://plos.org/protocols?utm_medium=editorial-email&utm_source=authorletters&utm_campaign=protocols.

We look forward to receiving your revised manuscript.

Kind regards,

Kimberly Page, PhD, MPH

Academic Editor

PLOS ONE

Journal Requirements:

2. Please include your ethical information in the Methods section of your manuscript

3. Please amend your current ethics statement to address the following concerns:

a) Did participants provide their written or verbal informed consent to participate in this study?

I have read the journal's policy and the authors of this manuscript have the following competing interests: Dawn Fishbein, MD receives grant funding for this research from Gilead Sciences, Inc through her employer. She has stock ownership in Abbvie and Merck is a discretionary portfolio with a value less than $10,000.

Reviewers' comments:

Reviewer's Responses to Questions

**Comments to the Author**

1. Is the manuscript technically sound, and do the data support the conclusions?

Reviewer #1: Yes

2. Has the statistical analysis been performed appropriately and rigorously? 

Reviewer #1: Yes

3. Have the authors made all data underlying the findings in their manuscript fully available?

Reviewer #1: Yes

4. Is the manuscript presented in an intelligible fashion and written in standard English?

Reviewer #1: Yes

5. Review Comments to the Author

Reviewer #1: In the paper, Hack and colleagues present an analysis of HCV screening practices before and after initiation of a BPA in an academic health system. They report that screening rates went up after the BPA was implemented and screening improved in adult groups who were the focus on new universal screening recommendations. The results are straightforward and support the use of a BPA to increase screening. I think the paper would benefit from some added information and discussion to help readers put these results into the proper context.

First, the readers needs more information about the health system mentioned. All outpatient visits are not the same, so some broad breakdown of visits for primary care, specialty care or ED would be helpful. This is also germane to the subgroups the authors mention with disparities, as some specialty physicians may not take ownership of screening when they see this as the purview of the primary care physicians. Also, a general breakdown of patients seen within the system (race, ethnicity, insurance breakdown) would be helpful to better understand how the screened population differs from all patients seen.

Second, there is no discussion of what education and/or preparation was provided prior to initiation. Were providers given a primer as to the policy changes that led to the BPA? Some physicians are very aware while others may be oblivious. We need more information to help understand the impact of these changes.

The authors state that more women were screened. Given that universal screening in pregnancy was a key inclusion in both public health recommendations, with the CDC recommending screening with every pregnancy, this is expected. The authors neglect to mention this subgroup in their results and it is a significant omission. Some additional discussion of pregnancy screening needs to be included.

The paper would benefit from some discussion about how the authors are planning to improve the use and function of the BPA. Aside from drawing attention to the disparate practices in screening different groups, how do we get to 100%?

6. PLOS authors have the option to publish the peer review history of their article (what does this mean?). If published, this will include your full peer review and any attached files.

Reviewer #1: No

---

## [Author Response · Author response to Decision Letter 0]

13 Dec 2022

See attached document with Response to Reviewers.

---

## [Editor Report · Decision Letter 1]

19 Dec 2022

HCV Universal EHR Prompt Successfully Increases Screening, Highlights Potential Disparities

PONE-D-22-18564R1

Dear Dr. Anzalone,

We’re pleased to inform you that your manuscript has been judged scientifically suitable for publication and will be formally accepted for publication once it meets all outstanding technical requirements.

Kind regards,

Kimberly Page, PhD, MPH

Academic Editor

PLOS ONE
---

## [Editor Report · Acceptance letter]

8 Feb 2023

PONE-D-22-18564R1 

HCV universal EHR prompt successfully increases screening, highlights potential disparities 

Dear Dr. Hack:

I'm pleased to inform you that your manuscript has been deemed suitable for publication in PLOS ONE. Congratulations! Your manuscript is now with our production department. 

Kind regards, 

on behalf of

Dr. Kimberly Page 

Academic Editor

PLOS ONE